# Antimicrobial Resistance in Vaginal Bacteria in Inseminated Mares

**DOI:** 10.3390/pathogens12030375

**Published:** 2023-02-24

**Authors:** Pongpreecha Malaluang, Elin Wilén, Sara Frosth, Johanna F. Lindahl, Ingrid Hansson, Jane M. Morrell

**Affiliations:** 1Department of Clinical Sciences, Swedish University of Agricultural Sciences (SLU), 75007 Uppsala, Sweden; 2Faculty of Veterinary Sciences, Mahasarakham University, Maha Sarakham 40000, Thailand; 3Evidensia Horse Clinic, 96174 Boden, Sweden; 4Biomedical Science and Veterinary Public Health, Swedish University of Agricultural Sciences (SLU), 75007 Uppsala, Sweden

**Keywords:** AMR, vaginal flora, horse breeding, semen extenders, resistance genes

## Abstract

Antimicrobials are added to semen extenders to inhibit the growth of bacteria that are transferred to the semen during collection. However, this non-therapeutic use of antimicrobials could contribute to the development of antimicrobial resistance. The objective of this study was to determine changes in the antibiotic susceptibility of vaginal microbiota after artificial insemination. Swabs were taken from the vagina of 26 mares immediately before artificial insemination and again 3 days later. Bacteria isolated from the vagina at both time points were subjected to antibiotic susceptibility testing and whole-genome sequencing. In total, 32 bacterial species were identified. There were increases in the resistance of *Escherichia coli* to trimethoprim (*p* = 0.0006), chloramphenicol and (*p* = 0.012) tetracycline (*p* = 0.03) between day 0 and day 3. However, there was no significant effect of exposure to antibiotics in semen extenders with respect to the resistance of *Staphylococcus simulans* and *Streptococcus equisimilis* (*p* > 0.05). Whole-genome sequencing indicated that most phenotypic resistance was associated with genes for resistance. These results indicate that the resistance patterns of vaginal bacteria may be affected by exposure to antibiotics; therefore, it would be prudent to minimize, or preferably, avoid using antibiotics in semen extenders.

## 1. Introduction

Antimicrobial resistance (AMR) has become a serious global concern that is accelerated by the misuse and overuse of antibiotics, as well as by inadequate infection prevention and control [1]. Although efforts are being made to reduce non-therapeutic uses of antimicrobials, one major use of antibiotics is still in animal breeding. The reproductive biotechnology artificial insemination (AI) was developed originally to prevent disease transmission when animals met for breeding, but it also facilitated faster genetic gain than could be achieved with natural mating. It is now a global phenomenon with millions of semen doses being traded around the world each year. For this international trade in semen, most countries have strict requirements regarding the health of semen donors, and they also stipulate the antibiotics that will be included in semen extenders. These antimicrobials are aimed at preventing the spread of bacterial diseases to inseminated females and the deterioration of sperm quality during storage [2].

There are several reports (reviewed recently [3]) concerning the AMR of bacteria isolated from uterine swabs, uterine lavage, vaginal swabs and clitoral swabs from the equine reproductive tract, mostly in cases of clinical disease or fertility problems. Briefly, resistance among isolates from the reproductive tract of mares has been reported in several countries including France [4], Sweden [5], India [6,7], Italy [8,9,10], Germany [11], the US [12,13], Slovakia [14] and Turkey [15]. Few detailed studies have been conducted on AMR in the reproductive microflora in healthy horses, and no studies have been conducted to investigate the effect of antimicrobial substances in semen extenders on vaginal flora. The purpose of this study, therefore, was to investigate whether the inclusion of antimicrobial substances in semen extenders affects the AMR patterns of the equine vaginal microbiota by conducting analyses of bacteria sampled before and after insemination.

## 2. Materials and Methods

### 2.1. Study Period

The study was conducted in Sweden during May–August 2020.

### 2.2. Mares

The mares (*n* = 26) included in the study were of different breeds and housed at the same stud farm in Boden, northern Sweden. Breeds included Swedish standardbred trotters, North Swedish Trotters, Arabian thoroughbred, Holstein and pony breeds. Their age varied from 5 to 20 years (average age 9 years); 22 mares had previously had foals (between 1 and 12 foals), and the remaining 4 mares were maiden mares. The inclusion criterion for the mares in this study was that they should not have previously been exposed to semen extenders via insemination during the breeding season of the year 2020.

Ethical approval for swabbing was available prior to the study (number 5.8.18-15533/2018).

### 2.3. Sampling Technique

Swabbing was conducted twice, once on day 0 (D0) immediately prior to insemination and the second time on day 3 (D3). Thus, each mare served as its own control. This sampling regimen was chosen as the most optimal to observe changes based on a pilot study [16]. For mares with a foal at foot, the D0 sample was taken in association with insemination, which took place at foal heat, approximately 7–10 days postpartum. In the case of a repeated insemination in a subsequent cycle, the swabs used were from D0 in the first insemination cycle and from D3 in the last insemination cycle.

The sampling was performed with the mare in an examination stock; the tail was wrapped before the mare’s perineal area and vulva were cleaned to remove all visible dirt, with at least three washes of soap and clean lukewarm water. The vulva and surrounding skin were then dried with bleached paper, which was inspected to determine the cleanliness of the prepared area. If necessary, further washing was carried out to achieve full cleanliness. The mares were sampled under a high hygienic standard, and a sterile glove over a rectal glove was used, avoiding touching the labia and using sterile liquid paraffin and a double-guarded occluded swab. The samples were taken from an area of the cranial vagina, approximately three centimeters distal to the fornix vagina; the precise anatomical area had been chosen after a previous inspection of organs from the slaughterhouse. The sampling site was identified by palpating the cervix uteri with a finger and then withdrawing by approximately one finger joint length. The sterile swab was in contact with the ventral vaginal wall for about 15 s while being gently rotated to collect an adequate sample.

The sampling swabs were then transferred directly into Amie’s transport medium, with charcoal (Copan Diagnostics, Inc. Murrieta, CA, USA) and stored at refrigerator temperature to prevent bacterial overgrowth. Samples were sent to the laboratory at the Swedish University of Agricultural Sciences (SLU) from Monday to Wednesday to reduce the risk of sitting at the post-terminal over the weekend.

### 2.4. Bacteriological Analyses

According to a previous study [16], various culture media were selected to increase the chances of bacterial growth based on bacteria detected in the vagina of other animal species. The sampling swab was cultured directly on cattle or horse blood agar plates (SVA, Uppsala, Sweden) for aerobic and anaerobic conditions; a direct culture was also performed on lactose purple agar (SVA, Uppsala, Sweden), MacConkey agar (SVA, Uppsala, Sweden) and Baird-Parker agar (Oxoid, Basingstoke, UK), which were incubated aerobically at 37 ± 1 °C. All plates incubated at 37 ± 1 °C were examined for bacterial growth after 24 and 48 h. For analyses of presumptive *Lactobacillus* spp., the swabs were cultured in De Man, Rosa and Sharpe agars (MRS-agar) (Oxoid, Basingstoke, UK), which were incubated anaerobically at 25 ± 1 °C for 5 days. Bacterial colonies of different macromorphologies from the initial culture were re-cultured on two blood agar plates (for aerobic and anaerobic culture conditions) and incubated for 24 to 48 h at 37 ± 1 °C to obtain a pure culture. The colonies from the pure cultures were stored in cryotubes with brain heart infusion (BHI) broth (CM1135; Oxoid, Basingstoke, UK) with 15% glycerol at −70 °C for subsequent antimicrobial-resistance testing and whole-genome sequencing.

The bacteria were identified at the species level by the use of matrix-assisted laser desorption ionization–time of flight mass spectrometry (MALDI-TOF MS) (Bruker Daltonics, Billerica, MA, USA). The mass spectrum of bacterial isolates was compared with those of known bacterial strains in the database (Bruker Daltonics, Billerica, MA, USA). Score values between 2.0 and 3.0 were considered accurate at both genus and species levels, whereas score values between 1.7 and 2.0 were considered reliable only at the genus level.

### 2.5. Antimicrobial Susceptibility Testing

The resistance to selected antimicrobial substances was determined for most commonly isolated bacteria present in samples both before (D0) and after insemination (D3) in the same mare. The isolates were cultured on horse blood agar directly from the cryotubes and incubated for 24 h at 37 °C. They were re-cultured once to ensure a pure culture.

The resistance panel used for *Staphylococcus* spp. and *Streptococcus* spp. was *Thermo Scientific™ Sensititre ™ STAFSTR*; for *Enterococcus faecalis*, it was Thermo Scientific™ Sensititre™ EUVENC; for *E. coli*, it was Thermo Scientific™ Sensititre™ EUVENSEC (Thermo Fisher Scientific, Waltham, Massachusetts, USA). Susceptibility to selected antimicrobial substances was assessed with VetMIC™ panel analysis systems: Camp EU, version 2013-10 (SVA, Uppsala, Sweden), determining the antimicrobial minimum inhibitory concentration (MIC) by broth microdilution following the standards of the Clinical and Laboratory Standards Institute [17].

A purity check and testing of concentration were performed on the culture of the final inoculum after placing the inoculum in the resistance panel on the horse blood agar, which was incubated at 37 °C for 24 h. Epidemiological cut-off (ECOFF) values for determining susceptibility were obtained from the European Committee on Antimicrobial Susceptibility Testing (EUCAST, www.eucast.org/mic_distributions_and_ecoffs, accessed on 20 February 2023). The ECOFF values classify isolates with acquired reduced susceptibility as “non-wild type” In this paper, non-wild-type isolates are called “resistant”, in agreement with the Swedish Veterinary Antibiotic Resistance Monitoring report [18].

### 2.6. Whole-Genome Sequencing

Whole-genome sequencing (WGS) was performed to determine whether the bacterial isolates of different antimicrobial susceptibilities from the same mares sampled at both time points were likely to be of the same strain or not and if phenotypic resistance was associated with genes for resistance. Resistant bacteria were subjected to WGS, including *E. coli, Streptococcus equisimilis* and *Staphylococcus simulans*. All isolates from the individual mares with at least one isolate phenotypically resistant to any antibiotics at both points were selected for WGS. WGS was performed on 127 *E. coli,* 26 *Streptococcus equisimilis* and 4 *Staphylococcus simulans* isolates, which showed increased or decreased resistance to antibiotics according to antimicrobial susceptibility testing. “Increased resistance” means that the proportion of bacteria from an individual mare showing resistance was higher later in the sampling sequence, whereas “decreased resistance “means that the proportion of bacteria showing resistance was lower later in the sampling sequence.

Isolates were subcultured twice on horse blood agar plates and from a single colony prior to DNA extraction to ensure pure culture. DNA was extracted with the EZ1 DNA Tissue Kit (Qiagen, Hilden, Germany) according to the protocols for Gram-negative and Gram-positive bacteria, respectively, with the exception that 75U of mutanolysin (Sigma–Aldrich, St. Louis, MO, USA) was added, and the lysis time extended to 4 h for *S. equisimilis.* The extraction was performed on Qiagen EZ1 Advanced XL utilizing the bacterial protocol. The elution volume used was 100 µL. The Qubit ds DNA HS kit (Invitrogen, Carlsbad, CA, USA) was used to measure DNA concentrations. 

Sequencing libraries were constructed with Nextera XT DNA Library Preparation Kit (Illumina, San Diego, CA, USA), and the quality was assessed by the HS DNA ScreenTape Analysis D1000 (Agilent Technologies, Inc., Santa Clara, CA, USA) on a 4150 TapeStation (Agilent Technologies, Inc.). Libraries were quantified by Qubit ds DNA HS kit (Invitrogen). WGS of the prepared libraries was conducted on the Illumina NextSeq 500 system (Illumina Inc., San Diego, CA, USA) using the Mid Output kit V2.5 with 2 × 150-bp paired-end reads (Illumina Inc).

Generated sequence reads were analyzed with SeqSphere + v7.0.5 software (Ridom GmbH, Münster, Germany). Genome assembly was done *de novo* by SKESA [19] via scripts in SeqSphere+ (Ridom GmbH). Multilocus sequencing typing (MLST) profiles were allocated for *E. coli* and *S. dysgalactiae*, respectively, using available schemes at http://enterobase.warwick.ac.uk/ [20] and https://pubmlst.org/ [21] via the MLST task templates in SeqSphere+ (Ridom, GmbH). The core genome MLST (cgMLST) for *E. coli* was performed with a task template containing 2513 loci in SeqSphere+ (Ridom, GmbH). For *S. dysgalactiae* and *S. simulans*, cgMLST schemes were created in SeqSphere+ (Ridom, GmbH). For *S. dysgalactiae*, NZ_LR594046.1 was used as the seed genome, and the following genomes were used as penetration query genomes: NZ_AP018726.1, NZ_CP044102.1, NZ_CP066073.1, NZ_CP066069.1, NZ_CP068057.1, NZ_CP033391.1, NZ_CP068478.1, JAAACI000000000.1, JAAACJ000000000.1, JAAACK000000000.1 and JAAACO000000000.1. The developed cgMLST scheme for *S. dysgalactiae* contained 1133 targets in total. For *S. simulans*, NZ_LS483313.1 was used as the seed genome, and NZ_CP014016.2, NZ_CP023497.1, NZ_CP017428.1, NZ_CP017430.1, NZ_CP016157.1, NZ_CP015642.1, NZ_LR134264.1, NZ_LT963435.1, AGZX00000000.1, LRQJ00000000.1, and PPRU00000000.1 were used as penetration query genomes. The final cgMLST scheme for *S. simulans* consisted of 1979 targets. Minimum spanning trees (MST: s) based on cgMLST data were created in SeqSphere+ (Ridom, GmbH) to investigate genetic relationship between isolates, and missing alleles were ignored in pairwise comparisons. For *E. coli*, a cluster distance threshold of 10 cgMLST targets was used.

Assembled genomes were screened for genes and point mutations associated with antimicrobial resistance by AMRFinderPlus [22] via SeqSphere+ (Ridom, GmbH), and ResFinder 4.1 [23,24,25].

### 2.7. Statistical Analysis

The difference between antimicrobial resistance results from vaginal isolates before and after contact with semen extenders was analyzed using the chi-squared test. If the criteria for the chi-squared test were not fulfilled, a Fisher’s exact test was used instead. A probability level of *p* < 0.05 was considered statistically significant.

## 3. Results

### 3.1. Bacterial Analyses

In total, 985 isolates of 32 different bacterial species were identified from the 26 mares. *E. coli* was the most common bacterial species isolated from 23 of 26 mares. The second and third most-isolated bacterial genera were *Streptococcus* spp. and *Acinetobacter* spp., which were identified in 24 of 26 mares and 17 of 26 mares, respectively (Table 1).

### 3.2. Antimicrobial Susceptibility

The resistance of *Escherichia coli* against trimethoprim (*p* = 0.0006), chloramphenicol (*p* = 0.012) and tetracycline (*p* = 0.03) increased from D0 to D3. However, *E. coli* resistance to other antibiotics was not different after exposure to semen extenders. Furthermore, there was no association between exposure to semen extenders and the resistance of *Staphylococcus simulans* and *Streptococcus equisimilis* (*p* > 0.05).

Resistance to at least 1 of 14 antibiotics was found in 284 of 520 *E. coli* isolates. Twenty-nine isolates were multidrug-resistant (resistant to at least three classes of antibiotics). There was a significant increase in resistance to trimethoprim, chloramphenicol and tetracycline after exposure to semen extenders with antibiotics. However, resistance was also found against sulfamethoxazole, ampicillin, azithromycin and tigecycline, but differences between time points were not significant. All *E. coli* isolates were susceptible to cefotaxime, ceftazidime, ciprofloxacin, colistin, gentamicin, meropenem and nalidixic acid (Table 2).

No correlation was found between the resistance of *Streptococcus dysgalactiae* subsp. *equisimilis* before and after exposure to semen extenders with antibiotics. Various *Streptococcus dysgalactiae* subsp. *equisimilis* isolates were resistant to three of eight antibiotics, including tetracycline, erythromycin and nitrofurantoin. All *Streptococcus dysgalactiae* subsp. *equisimilis* isolates were susceptible to cefalothin, clindamycin, oxacillin, penicillin and trimethoprim/sulfamethoxazole. For some antibiotics, cefoxitin, enrofloxacin, fusidic acid and gentamicin, sensitivity or resistance could not be assessed since no ECOFFs were mentioned in EUCAST (Table 3).

All *Enterococcus faecalis* isolates were susceptible to the six tested antibiotics (ampicillin, ciprofloxacin, gentamicin, linezolid, teicoplanin and tigecycline). For some antibiotics (chloramphenicol, daptomycin, erythromycin, quinupristin/dalfopristin, tetracycline and vancomycin), sensitivity or resistance could not be assessed since no ECOFFs were given in the EUCAST (Table 4). All tested *Streptococcus equi* subsp. *zooepidemicus* isolates were susceptible to clindamycin, erythromycin, nitrofurantoin, oxacillin, penicillin, tetracycline and trimethoprim/sulfamethoxazole. For some antibiotics, cefalotin, cefoxitin, enrofloxacin, fusidic acid and gentamicin sensitivity or resistance could not be assessed since no ECOFFs were given in the EUCAST (Table 5).

Resistance to at least one of seven antibiotics was detected in different *Staphylococcus simulans* isolates, including oxacillin (25%), penicillin (50%), fusidic acid (50%) and trimethoprim/sulfamethoxazole (50%). One isolate was multidrug resistant. There was no association between exposure to antibiotics via artificial insemination and change in resistance. All *Staphylococcus simulans* isolates were susceptible to erythromycin, gentamicin and tetracycline. For some antibiotics, cefalotin, cefoxitin, clindamycin, enrofloxacin and nitrofurantoin, sensitivity or resistance could not be assessed since no ECOFFs were given in the EUCAST (Table 6).

### 3.3. Whole-Genome Sequencing

Whole-Genome Sequencing was used to decide whether *E. coli*, *Streptococcus equisimilis* and *Staphylococcus simulans* isolates of different antimicrobial susceptibility from the same mares sampled at both time points were likely to be of the same strain or not and if phenotypic resistance was associated with genes for resistance.

One hundred twenty-seven *E. coli* isolates from inseminated mares before (D0) and after (D3) insemination, were allocated into fourteen clusters by cgMLST analysis (Figure 1). There was, at most, a one-allele difference between isolates within each cluster, except for Cluster 2 and 13 where there was a three-allele difference. Therefore, isolates within each cluster were highly genetically related, i.e. the same strain. Nine clusters consisted of isolates from one mare (Cluster 3, 6, 7, 8, 9, 10, 11 and 14), whereas five clusters consisted of isolates from 2 to 3 mares (Cluster 1, 2, 4 5 and 13 (Figure 1)). The finding of isolates from different mares in the same clusters suggests a possible spread within the herd.

Twenty-six *Streptococcus equisimilis* isolates from inseminated mares before (D0) and after (D3) insemination, were subjected to cgMLST analysis, and a minimum spanning tree (MST) was generated (Figure 2a). There were 18 allelic differences between 12 isolates from Horse H and 3 isolates from Horse J, but since the developed cgMLST scheme lacked a validated clustering distance threshold, it was not possible to determine whether the isolates belonged to 2 different clusters or 1 and the same (Figure 2a). Regardless, there were no allelic differences between isolates from each horse in this case; hence, they were of the same strain. Eleven isolates from Horse I and J clustered together with a maximum of two allelic differences, and even if the developed cgMLST scheme lacked a validated clustering distance threshold, it is likely that those isolates were genetically closely related; i.e. they were the same strain since the differences were very few (Figure 2a). The finding of genetically related isolates in two mares (Horse I and J) indicates a possible spread within the herd, as seen in *E. coli* isolates.

Four *Staphylococcus simulans* isolates from an inseminated mare (Horse K) before (D0) and after (D3) insemination, were subjected to cgMLST analysis, and a minimum spanning tree (MST) was generated (Figure 2b). There were no allelic differences between isolates P868 and PM194 and P869 and PM193, respectively; hence, isolates within each pair were genetically closely related, i.e. the same strain. Although no defined cluster distance threshold was available for this cgMLST scheme, the allelic distance between the two pairs was very large, and it is therefore unlikely that they are genetically closely related (Figure 2b).

Twenty-nine of the *E. coli* isolates were multiple drug resistant (MDR); twenty of these isolates belonged to Cluster 2 (Horse C and G), with one isolate each belonging to Cluster 4 (Horse B), Cluster 6 (Horse E), Cluster 7 (Horse D) and Cluster 14 (Horse B).

Eighteen of the *Streptococcus equisimilis* isolates were phenotypically resistant to tetracycline; eight clustered together from Horse H; ten clustered with isolates from Horse I and J. One isolate from Horse H was phenotypically resistant to erythromycin. One isolate was phenotypically resistant to nitrofurantoin, and it clustered together with isolates from Horse I and J.

Two isolates (P868 and PM194), which clustered together, were phenotypically resistant to fusidic acid. Two other isolates that clustered together (P869 and PM193) were phenotypically resistant to penicillin and trimethoprim/sulfamethoxazole. One isolate (P869) was phenotypically resistant to oxacillin.

The phenotypic resistance was compared between isolates from individual mares before and after the exposure to antibiotics within the same cluster, i.e. isolates of the same strain (Table 7 and Table 8). For *E. coli,* increased resistance to sulfamethoxazole (Cluster 1), trimethoprim (Cluster 2 and 10), chloramphenicol (Cluster 2, 6 and 14) and tetracycline (Cluster 6) was found.

#### AMR Genes

All sequenced *Escherichia coli* isolates were resistant to sulfamethoxazole both phenotypically and genotypically; 6 isolates had the *sul1* gene, and 26 isolates had the *sul2* gene, which are responsible for sulfamethoxazole resistance. However, no resistance genes were found in 95 of the isolates. Thirty-two *E. coli* isolates were phenotypically resistant to trimethoprim; twenty-six of these isolates had *dfrA1*; seventeen isolates had *dfrA14;* no resistance genes were found in two isolates. Ten *E. coli* isolates were phenotypically resistant to tetracycline; six of these isolates had *tet(A)*; one isolate had *marR_S3N*; no resistance genes were found in the other three isolates. All 32 *E. coli* isolates that were phenotypically resistant to chloramphenicol had *mdf(A)*. One *E. coli* isolate phenotypically resistant to tigecycline had *mdf(A)*. Of the seven *E. coli* isolates phenotypically resistant to ampicillin, six isolates had *blaEC-5/blaTEM-1* and *blaTEM-1B*; one isolate had *blaEC*.

The gene *marR_S3N* was found in twenty-seven *E. coli* isolates susceptible to ciprofloxacin, chloramphenicol and tigecycline and twenty-six isolates susceptible to tetracycline. The gene *blaEC* was found in 119 isolates susceptible to meropenem, cefotaxime and ceftazidime and 118 isolates susceptible to ampicillin. The gene *blaEC-5*/*blaTEM-1* was found in six isolates susceptible to meropenem, cefotaxime and ceftazidime.

The gene *catB3* was found in six *E. coli* isolates susceptible to chloramphenicol. The gene *tet(A)* was found in six isolates susceptible to tigecycline. The gene *aadA5* was found in six isolates susceptible to gentamicin. The gene *aph(6)-ld* was found in 26 isolates susceptible to gentamicin. In addition, the gene *mdf(A)*, conferring resistance to broad-spectrum drugs via an efflux pump, was found in 92 isolates.

Five *Streptococcus equisimilis* isolates had *lsaC* as a resistance gene; one of these isolates had phenotypical resistance to erythromycin. Eighteen isolates phenotypically resistant to tetracycline were not shown to have resistance genes. Furthermore, one isolate, which was phenotypically resistant to nitrofurantoin, did not have resistance genes.

Two *Staphylococcus simulans* isolates had the *blaZ* resistance gene responsible for phenotypical resistance to penicillin and oxacillin. However, two isolates phenotypically resistant to fusidic acid and two isolates phenotypically resistant to trimethoprim/sulfamethoxazole did not have any resistance genes.

## 4. Discussion

The purpose of this study was to determine if the vaginal microbiota of mares was affected by exposure to antibiotics in semen extenders and also whether this exposure affected the AMR of the bacteria isolated from the vagina.

### 4.1. Bacterial Analyses

In the present study, 300 Gram-positive and 685 Gram-negative bacterial isolates were identified from the vagina. This result is in agreement with other studies where Gram-negative bacteria dominated the microbiota in the vagina [7,26]. The isolates most frequently found in our study included *E. coli* (52.8%), *S. equisimilis* (14.0%) and *S. zooepidemicus* (3.6%). A study in Korea revealed that the most frequently isolated species of bacteria were *Escherichia coli* (19.8%), *Staphylococcus aureus* (14.9%) and *Proteus mirabillis* (14.9%)*;* the samples were collected from vaginal mucosa and clitoral fossa [26]. In contrast, in an Indian report, the dominating bacteria in the vagina were *E. coli* (21.7%), *Enterobacter agglomerans* (16.7%)*, Enterococcus faecalis* (15.6%) and *Enterococcus faecium* (15.6%) [7]. However, bacteria in these other studies were isolated using various types of agar plates and cultured under different conditions compared to our study, and they were identified at the species level using different methods compared to ours. Although other methods of analysis were used in different studies, *E. coli* seems to be the most isolated bacterial species, most likely because it is one of the most common bacteria in the reproductive tract and the surroundings.

*Lactobacillus* spp. were not detected in our study, although the selective culture medium, MRS agar, was used. This observation is in contrast to a study performed in Uruguay, in which *Lactobacillus* spp. were isolated from all vaginal samples [27]. Time and temperature might affect the viability of *Lactobacilli*; in the study by Fraga et al. [27], the swabs were kept at 4 °C and transported to the laboratory within 3 h. In our study, the stud was at a considerable distance (> 860 km) from the laboratory, necessitating overnight transport of the swabs, which took place at ambient temperatures.

### 4.2. Antimicrobial Susceptibility

The vaginal microbiota of mares in our study was susceptible to most of the tested antibiotics. In fact, all *Enterococcus faecalis* isolates were susceptible to all 12 tested antibiotics in our study. In other studies, most isolates were obtained from infertile mares, which may have been treated with antibiotics. In other studies, *Enterococcus* spp. isolates have been described as resistant to all antibiotics tested [7] and resistant to some antibiotics except penicillin, cefquinome, florfenicol, amoxicillin/clavulanic acid, amoxicillin, gentamicin and colistin [15] or resistant to some antibiotics except for gentamicin, vancomycin, linezolid, penicillin and tetracycline [9].

Interestingly, all *Streptococcus equi* subsp. *zooepidemicus* isolates of the exposed mares were susceptible to all tested antibiotics. This result is in line with monitoring antibiotic resistance in veterinary medicine in Sweden, where *Streptococcus zooepidemicus* isolated from horses have remained uniformly susceptible over the years [18]. In contrast, studies of this bacterium from other countries reported resistance to most antibiotics [10,12,28].

Resistance to seven different antibiotics was found in different *E. coli* isolates in our study, including sulfamethoxazole (90%), trimethoprim (10.7%), chloramphenicol (10.6%), and tetracycline (2%) ampicillin (2.5%), azithromycin (0.5%) and tigecycline (0.5%). Trimethoprim, chloramphenicol and tetracycline showed increased resistance three days after AI. However, these antibiotics were not included in the semen extender. Genes conferring resistance to antibiotics may not be specific for these antibiotics but ‘merely’ allow survival in a hostile environment, with resistance to antibiotics being coincidental. Therefore, bacteria do not necessarily need to be exposed to an antibiotic to develop resistance to it. Alternatively, exposure to another antibiotic could cause a transfer of resistance between bacteria, e.g. via plasmids [29]. Other factors might influence the association, such as previous treatment or exposure to environmental bacteria on personnel. The resistance of *E. coli* isolates to some tested antibiotics in our study was similar to other studies, where trimethoprim, chloramphenicol, tetracycline, ampicillin and azithromycin resistance was found [11,13,15]. To our knowledge, there have not been any previous reports of resistance to tigecycline in *E. coli* isolates from the vagina of mares. Resistance to 4 of 14 antibiotics could be detected in different *Staphylococcus simulans* isolates in our study, including oxacillin (25%), penicillin (50%), fusidic acid (50%) and trimethoprim/sulfamethoxazole (50%). Other studies showed the resistance of *Staphylococcus* spp. to most antibiotics tested, including amoxicillin/clavulanic acid, cefotaxime, cefquinome, colistin, enrofloxacin, erythromycin, gentamicin, kanamycin, marbofloxacin, penicillin, rifampicin, sulfisoxazole, tetracycline and trimethoprim/sulfamethoxazole [8,14,15]. As in our study, penicillin and trimethoprim/sulfamethoxazole resistance were found in these studies.

Many isolates (91.1%) of *Streptococcus equisimilis* were resistant to tetracycline but showed low resistance to erythromycin (1.1%) and nitrofurantoin (1.1%). Furthermore, all *Streptococcus equisimilis* isolates in this study were susceptible to cefalotin, clindamycin, oxacillin, penicillin and trimethoprim/sulfamethoxazole, which differed significantly from some other studies where resistance to all tested antibiotics was reported [7,8].

### 4.3. Whole-Genome Sequencing

Following antimicrobial resistance testing results, sequenced *E. coli* isolates from inseminated mares at both time points were separated into 14 MST clusters; 9 clusters consisted of isolates from 1 mare, whereas 5 clusters consisted of isolates from 2 to 3 mares. The finding of genetically related isolates in more than one mare indicates a possible spread within the herd. Moreover, for *Streptococcus equisimilis*, there was an indication of spread within the herd, since the same isolate was potentially detected in isolates from two horses (Horse I and J). Twenty-nine of the sequenced *E. coli* isolates (22.8%) were MDR, and they were found in five clusters; one cluster consisted of isolates from two mares. The possibility of spreading within the herd might occur also for MDR isolates.

Increased or decreased resistance was observed from isolates from the same horse within the same cluster. Following antimicrobial resistance testing results, the proportions of resistant isolates before and after insemination were compared. Increased resistance to sulfamethoxazole, trimethoprim, chloramphenicol and tetracycline and decreased resistance to azithromycin and tigecycline were found in *E. coli* isolates. *Streptococcus equisimilis* showed increased resistance to tetracycline and decreased resistance to erythromycin. An increase in phenotypic resistance might be caused by different mechanisms after experiencing an unfavorable environment; for example, it was reported previously that low-level exposure to antibiotics could lead to antimicrobial resistance [30]. The vaginal bacteria of the mares in this study were exposed to the antibiotics in the semen extender used to prepare the semen doses. However, since single isolates were selected for WGS based on microdilution results in our study, it is not possible to be certain that the same bacterial strain was chosen on all sampling occasions.

Two *E. coli* isolates and one *Streptococcus equisimilis* isolate, which had phenotypic resistance, did not contain resistance genes. Furthermore, there were no resistance genes in two *Staphylococcus simulans* isolates showing phenotypic resistance to fusidic acid and trimethoprim/sulfamethoxazole. Natural or acquired resistance usually results from resistance genes. A study showed that many resistance genes occurring in environmental bacteria might potentially be transferred to bacteria in the mares’ reproductive system [31]. Gene transmission in nature occurs via three mechanisms: conjugation, transduction and transformation [32]. Even dead bacteria can pass the resistance gene to other bacteria by transformation [33]. In the present study, possible explanations for the difference between phenotypic and genotypic resistance could be that the genes or the mechanisms conferring resistance have not been detected yet, or there may be a discrepancy between genotypic and phenotypic resistance.

In conclusion, most isolates from the cranial vagina of mares were Gram-negative bacteria. The exposure of vaginal bacteria to antibiotics in the semen extender was associated with a change in the resistance of *Escherichia coli* against trimethoprim, chloramphenicol and tetracycline. The antibiotic resistance pattern of the genetically related isolates in the same horse increased or decreased in some *Escherichia coli* and *Streptococcus equisimilis* isolates after exposure to the semen extender. These results are interesting and warrant further study.

## Figures and Tables

**Figure 1 pathogens-12-00375-f001:**
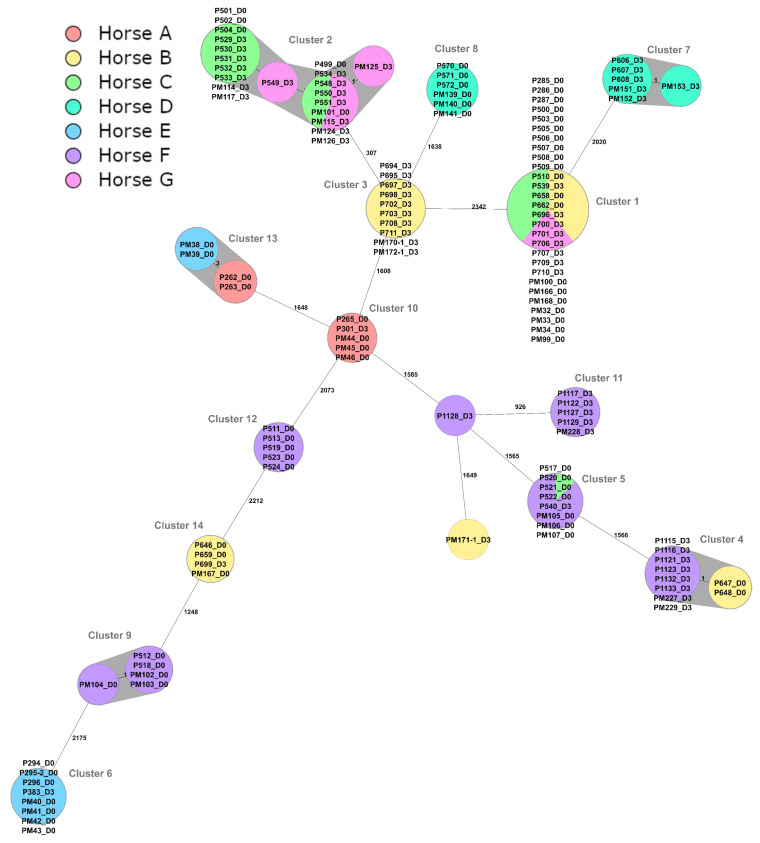
Minimum spanning tree (MST) created for 127 *Escherichia coli* from mares in Boden in northern Sweden visualizing core genome multi-locus sequence typing (cgMLST) results. Nodes corresponding to sequenced isolates are colored according to individual mares. D0 and D3 refer to isolates before and after insemination, respectively. The numbers between nodes represent allelic differences. Line lengths are not proportional to the number of differences. Identified clusters have been highlighted in grey.

**Figure 2 pathogens-12-00375-f002:**
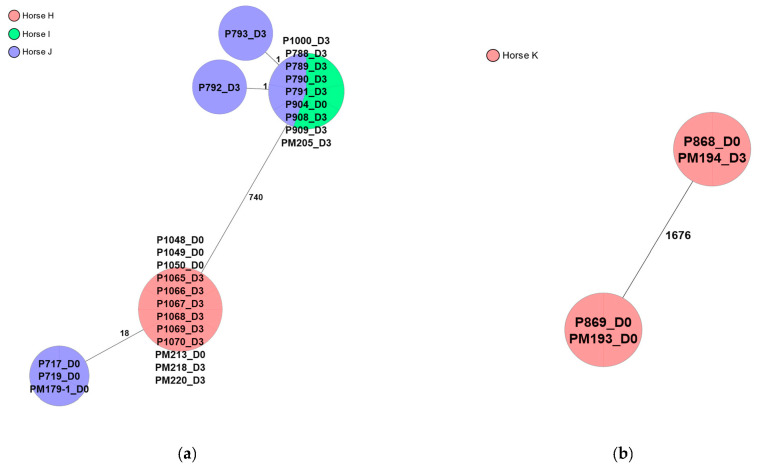
(**a**) Minimum spanning tree (MST) created for 26 *Streptococcus equisimilis* from mares in Boden in northern Sweden visualizing core genome multi-locus sequence typing (cgMLST) results. (**b**) Minimum spanning tree (MST) created for four *Staphylococcus simulans* from mares visualizing core genome multi-locus sequence typing (cgMLST) results. Nodes corresponding to sequenced isolates are colored according to individual mares. D0 and D3 are referring to isolates before and after insemination, respectively. The numbers between nodes represent allelic differences. Line lengths are not proportional to the number of differences.

**Table 1 pathogens-12-00375-t001:** Bacteria isolated from the cranial vagina before (D0) and after insemination (D3) from 26 Swedish mares. The percentage (%) is the percentage of all identified bacteria.

Bacteria	D0	D3	Total Number	(%)	No. of Mares
**Gram-Negative**					
*Acinetobacter bohemicus*	−	1	1	0.1	1
*Acinetobacter kookii*	−	2	2	0.2	1
*Acinetobacter lwoffii*	21	30	51	5.2	12
*Acinetobacter schindleri*	46	26	72	7.3	13
*Acinetobacter* spp.	2	−	2	0.2	2
*Escherichia coli*	179	341	520	52.8	23
*Klebsiella oxytoca*	−	21	21	2.1	1
*Pantoea agglomerans*	−	3	3	0.3	1
*Rahnella aquatilis*	1	1	2	0.2	2
*Serratia marcescens*	11	−	11	1.1	1
**Gram-Positive**					
*Aerococcus viridans*	6	−	6	0.6	1
*Arthrobacter gandavensis*	3	−	3	0.3	1
*Corynebacterium callunae*	−	5	5	0.5	2
*Corynebacterium casei*	3	−	3	0.3	1
*Enterococcus casseliflarus*	−	15	15	1.5	4
*Enterococcus faecalis*	6	40	46	4.7	4
*Enterococcus mundtii*	−	1	1	0.1	1
*Lactococcus raffnolactis*	−	2	2	0.2	1
*Paenibacillus amylolyticus*	−	2	2	0.2	2
*Staphylococcus aureus*	1	−	1	0.1	1
*Staphylococcus capitis*	4	5	9	0.9	3
*Staphylococcus haemolyticus*	−	2	2	0.2	2
*Staphylococcus schleiferi*	−	10	10	1.0	3
*Staphylococcus simulans*	3	1	4	0.4	1
*Staphylococcus vitulinus*	4	2	6	0.6	2
*Streptococcus canis*	−	2	2	0.2	2
*Streptococcus equisimilis*	48	90	138	14.0	18
*Streptococcus equinus*	1	3	4	0.4	2
*Streptococcus gallolyticus*	−	4	4	0.4	1
*Streptococcus hyovaginalis*	1	−	1	0.1	1
*Streptococcus thoraltensis*	1	−	1	0.1	1
*Streptococcus zooepidemicus*	8	27	35	3.6	9
Total	349	636	985	100.0	

**Table 2 pathogens-12-00375-t002:** Distribution of MICs (mg/L) and resistance of the 301 isolates of *Escherichia coli* from the cranial vagina of 14 mares sampled before (D0) and after (D3) insemination. The results are shown as the percentage of isolates at different MIC values.

		Res (%)	<0.12	0.25	0.5	1	2	4	8	16	32	64	>64
Ampicillin	D0 (*n* = 147)	1				4	33	59	3	1			
	D3 (*n* = 154)	4					21	72	3			4	
Azithromycin	D0 (*n* = 147)	1					13	51	33	2	1		
	D3 (*n* = 154)	0						38	58	4			
Cefotaxime	D0 (*n* = 147)	0		100									
	D3 (*n* = 154)	0		100									
Ceftazedime	D0 (*n* = 147)	0			99	1							
	D3 (*n* = 154)	0			99	1							
Ciprofloxacin	D0 (*n* = 147)	0	99	1									
	D3 (*n* = 154)	0	100										
Chloramphenicol *	D0 (*n* = 147)	6							94	5		1	
	D3 (*n* = 154)	15							85	14	1		
Colistin	D0 (*n* = 147)	0				99	1						
	D3 (*n* = 154)	0				100							
Gentamycin	D0 (*n* = 147)	0			73	26	1						
	D3 (*n* = 154	0			81	16	3						
Meropenem	D0 (*n* = 147)	0	100										
	D3 (*n* = 154)	0	100										
Nalidixic acid	D0 (*n* = 147)	0						99	1				
	D3 (*n* = 154)	0						99	1				
Sulfamethoxazole	D0 (*n* = 147)	90							1	1	3	5	90
	D3 (*n* = 154)	90								2	1	7	90
Tetracycline *	D0 (*n* = 147)	0					87	12	1				
	D3 (*n* = 154)	4					85	9	2	1	3		
Tigecycline	D0 (*n* = 147)	1		98	1			1					
	D3 (*n* = 154)	0		96	4								
Trimethoprim *	D0 (*n* = 147)	4		50	42	4					4		
	D3 (*n* = 154)	17		21	49	12	1				17		

White fields denote the range of dilutions tested for each antibiotic and vertical bold lines indicate cut-off values used to define resistance. Grey fields denote the range outside of dilutions tested. MICs equal to or lower than the lowest concentration tested are given as the lowest tested concentration. * Significant association between antibiotics exposure and resistance results (*p* < 0.05).

**Table 3 pathogens-12-00375-t003:** Distribution of MICs (mg/L) and resistance of the 90 isolates of *Streptococcus dysgalactiae* subsp. *Equisimilis* from the cranial vagina of 10 mares sampled before (D0) and after (D3) insemination. The results are shown as the percentage of isolates at different MIC values.

		Res (%)	<0.03	0.06	0.12	0.25	0.5	1	2	4	8	16	32	64	>64
Cefalotin	D0 (*n* = 48)	0						100							
	D3 (*n* = 42)	0						100							
Cefoxitin	D0 (*n* = 48)	−					17	83							
	D3 (*n* = 42)	−				2	10	88							
Clindamycin	D0 (*n* = 48)	0					100								
	D3 (*n* = 42)	0					100								
Enrofloxacin	D0 (*n* = 48)	−					15	85							
	D3 (*n* = 42)	−					21	79							
Erythromycin	D0 (*n* = 48)	2					98	2							
	D3 (*n* = 42)	0					100								
Fusidic acid	D0 (*n* = 48)	−							100						
	D3 (*n* = 42)	−							100						
Gentamicin	D0 (*n* = 48)	−							8	92					
	D3 (*n* = 42)	−							2	98					
Nitrofurantoin	D0 (*n* = 48)	0										81	19		
	D3 (*n* = 42)	2										84	14		2
Oxacillin	D0 (*n* = 48)	0				100									
	D3 (*n* = 42)	0				100									
Penicillin	D0 (*n* = 48)	0	100												
	D3 (*n* = 42)	0	100												
Tetracycline	D0 (*n* = 48)	87							13	87					
	D3 (*n* = 42)	95							5	95					
Trimethoprim/	D0 (*n* = 48)	0				100									
Sulfamethoxazole	D3 (*n* = 42)	0				98	2								

White fields denote the range of dilutions tested for each antibiotic, and vertical bold lines indicate cut-off values used to define resistance. Grey fields denote the range outside of dilutions tested. MICs equal to or lower than the lowest concentration tested are given as the lowest tested concentration.

**Table 4 pathogens-12-00375-t004:** Distribution of MICs (mg/L) and resistance of the 20 isolates of *Enterococcus faecalis* from the cranial vagina of 2 mares sampled before (D0) and after (D3) insemination. The results are shown as the percentage of isolates at different MIC values.

		Res (%)	<0.03	0.06	0.12	0.25	0.5	1	2	4	8	16	32	64	128	256	512	1024
Ampicillin	D0 (*n* = 6)	0						83	17									
	D3 (*n* = 14)	0						100										
Chloramphenicol	D0 (*n* = 6)	−									100							
	D3 (*n* = 14)	−									100							
Ciprofloxacin	D0 (*n* = 6)	0							100									
	D3 (*n* = 14)	0						21	79									
Daptomycin	D0 (*n* = 6)	−					33	17		50								
	D3 (*n* = 14)	−					7	72		14	7							
Erythromycin	D0 (*n* = 6)	−						50	50									
	D3 (*n* = 14)	−						36	64									
Gentamicin	D0 (*n* = 6)	0									83	17						
	D3 (*n* = 14)	0									93	7						
Linezolid	D0 (*n* = 6)	0							100									
	D3 (*n* = 14)	0							100									
Quinupristin/	D0 (*n* = 6)	−									50	50						
Dalfopristin	D3 (*n* = 14)	−									36	57	7					
Teicoplanin	D0 (*n* = 6)	0					100											
	D3 (*n* = 14)	0					100											
Tetracycline	D0 (*n* = 6)	−						100										
	D3 (*n* = 14)	−						100										
Tigecycline	D0 (*n* = 6)	0		100														
	D3 (*n* = 14)	0		86	14													
Vancomycin	D0 (*n* = 6)	−						83	17									
	D3 (*n* = 14)	−						64	29	7								

White fields denote the range of dilutions tested for each antibiotic, and vertical bold lines indicate cut-off values used to define resistance. Grey fields denote the range outside of dilutions tested. MICs equal to or lower than the lowest concentration tested are given as the lowest tested concentration.

**Table 5 pathogens-12-00375-t005:** Distribution of MICs (mg/L) and resistance of the eight isolates of *Streptococcus equi* subsp. *zooepidemicus* from the cranial vagina of two mares sampled before (D0) and after (D3) insemination. The results are shown as the percentage of isolates at different MIC values.

		Res (%)	<0.03	0.06	0.12	0.25	0.5	1	2	4	8	16	32	64	>64
Cefalotin	D0 (*n* = 5)	−						100							
	D3 (*n* = 3)	−						100							
Cefoxitin	D0 (*n* = 5)	−					60	40							
	D3 (*n* = 3)	−					67	33							
Clindamycin	D0 (*n* = 5)	0					100								
	D3 (*n* = 3)	0					100								
Enrofloxacin	D0 (*n* = 5)	−					80	20							
	D3 (*n* = 3)	−					100								
Erythromycin	D0 (*n* = 5)	0					100								
	D3 (*n* = 3)	0					100								
Fusidic acid	D0 (*n* = 5)	−							100						
	D3 (*n* = 3)	−							100						
Gentamicin	D0 (*n* = 5)	−						100							
	D3 (*n* = 3)	−						100							
Nitrofurantoin	D0 (*n* = 5)	0										100			
	D3 (*n* = 3)	0										100			
Oxacillin	D0 (*n* = 5)	0				100									
	D3 (*n* = 3)	0				100									
Penicillin	D0 (*n* = 5)	0	100												
	D3 (*n* = 3)	0	100												
Tetracycline	D0 (*n* = 5)	0							100						
	D3 (*n* = 3)	0							100						
Trimethoprim/	D0 (*n* = 5)	0				100									
Sulfamethoxazole	D3 (*n* = 3)	0				100									

White fields denote the range of dilutions tested for each antibiotic, and vertical bold lines indicate cut-off values used to define resistance. Grey fields denote the range outside of dilutions tested. MICs equal to or lower than the lowest concentration tested are given as the lowest tested concentration.

**Table 6 pathogens-12-00375-t006:** Distribution of MICs (mg/L) and resistance of the four isolates of *Staphylococcus simulans* from the cranial vagina of one mare sampled before (D0) and after (D3) insemination. The results are shown as the percentage of isolates at different MIC values.

		Res (%)	<0.03	0.06	0.12	0.25	0.5	1	2	4	8	16	32	64
Cefalotin	D0 (*n* = 3)	−						100						
	D3 (*n* = 1)	−						100						
Cefoxitin	D0 (*n* = 3)	−							67	33				
	D3 (*n* = 1)	−							100					
Clindamycin	D0 (*n* = 3)	−					100							
	D3 (*n* = 1)	−					100							
Enrofloxacin	D0 (*n* = 3)	−				100								
	D3 (*n* = 1)	−				100								
Erythromycin	D0 (*n* = 3)	0					100							
	D3 (*n* = 1)	0					100							
Fusidic acid	D0 (*n* = 3)	33						67	33					
	D3 (*n* = 1)	100							100					
Gentamicin	D0 (*n* = 3)	0						100						
	D3 (*n* = 1)	0						100						
Nitrofurantoin	D0 (*n* = 3)	−										100		
	D3 (*n* = 1)	−										100		
Oxacillin	D0 (*n* = 3)	33				67	33							
	D3 (*n* = 1)	0				100								
Penicillin	D0 (*n* = 3)	67	33					67						
	D3 (*n* = 1)	0	100											
Tetracycline	D0 (*n* = 3)	0						67	33					
	D3 (*n* = 1)	0						100						
Trimethoprim/	D0 (*n* = 3)	67					33			67				
Sulfamethoxazole	D3 (*n* = 1)	0				100								

White fields denote the range of dilutions tested for each antibiotic, and vertical bold lines indicate cut-off values used to define resistance. Grey fields denote the range outside of dilutions tested. MICs equal to or lower than the lowest concentration tested are given as the lowest tested concentration.

**Table 7 pathogens-12-00375-t007:** Distribution of resistant *E. coli* isolates and corresponding cluster and horse from which samples were originating.

Antibiotics (Total No. of Resistant Isolates).	No. of Resistant Isolates/Cluster/Horse
Trimethoprim (*n* = 32)	20/2/C; 2/4/B; 6/7/D; 4/10/A
Tetracycline (*n* = 10)	1/10/G; 2/2/G; 1/6/E; 6/7/D
Azithromycin (*n* = 1)	1/14/B
Chloramphenicol (*n* = 31)	20/2/C+G; 1/4/F; 1/5/F; 1/6/E; 1/8/D; 4/11/F; 3/14/B
Tigecycline (*n* = 1)	1/14/B
Ampicillin (*n* = 7)	1/4/B; 6/7/D

**Table 8 pathogens-12-00375-t008:** Phenotypic antimicrobial resistance of bacteria isolates from the vagina of mares before (D0) and after (D3) insemination.

Bacteria.	Cluster/ Horse	Day 0		Day 3	
		No. of Isolates	Resistance	No. of Isolates	Resistance
*E. coli*	1/C	10	Sul	1	Sul
	1/B	3	Sul	7	Sul
		1	−		
	2/C	4	Sul, Tri, Chl	9	Sul, Tri, Chl
		1	Sul		
	6/E	7	Sul	1	Sul, Tet, Chl
	10/A	1	Sul	4	Sul, Tri
	14/B	1	Sul, Azi, Chl, Tig		
		1	Sul, Chl	1	Sul, Chl
		1	Sul		
*S. equisimilis*	NA/I	1	−	4	Tet
	NA/H	2	Tet	8	Tet
		1	Ery		−
		1	−		
*S. simulans*	NA/K	1	Fus	1	Fus

Sul—sulfamethoxazole; Tet—tetracycline; Tri—trimethoprim; Chl—chloramphenicol; Azi—azithromycin; Tig—tigecycline; Ery—erythromycin; Fus—fusidic acid, NA—clustering is not available.

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
