# Peer review of "Antimicrobial Resistance in Vaginal Bacteria in Inseminated Mares"

_pathogens, 2023, doi:10.3390/pathogens12030375_

Round 1

Reviewer 1 Report

The manuscript entitled “Antimicrobial Resistance in Vaginal Bacteria in Inseminated Mares” investigates how resistance patterns of vaginal bacteria are affected by exposure of antibiotics.

The work face the global concern of antimicrobial resistance, an actual topic that is essential for human, animal and environmental health.

The topic is well developed starting from a well-organized introduction that describes the background and the purpose of the study.

Also the other sections are well structured making the whole manuscript fluid and interesting.

In my opinion, the manuscript can be accepted in its current form for the publication.

Regards

Reviewer 2 Report

The authors aimed to determine the changes in antibiotic susceptibility of vaginal microbiota after artificial insemination.

Since there is no control group, I suggest that the authors modify slightly the aim of the manuscript and do not make statistical analysis, and limit the findings to describing the dynamism of vaginal bacteria and resistance determinants (2-time points) among inseminated mares.

The authors should revise the name of antibiotics. Many are wrongly written (e.g., gentamycin should be gentamicin).

Overall, the manuscript is well-written and can be accepted after the suggested adjustments.

Round 2

Reviewer 2 Report

The manuscript can be accepted in its present form.